# Cryo-EM structures of adenosine receptor A₃AR bound to selective agonists

Hongmin Cai [1,7] ✉, Shimeng Guo [1,7], Youwei Xu [1,7], Jun Sun[1,2,7], Junrui Li[1], Zhikan Xia[1], Yi Jiang [3], Xin Xie [1,2,4,5,6] ✉ & H. Eric Xu [1,2,5] ✉

The adenosine A₃ receptor (A₃AR), a key member of the G protein-coupled receptor family, is a promising therapeutic target for inflammatory and cancerous conditions. The selective A₃AR agonists, CF101 and CF102, are clinically significant, yet their recognition mechanisms remained elusive. Here we report the cryogenic electron microscopy structures of the full-length human A₃AR bound to CF101 and CF102 with heterotrimeric Gᵢ protein in complex at 3.3-3.2 Å resolution. These agonists reside in the orthosteric pocket, forming conserved interactions via their adenine moieties, while their 3-iodobenzyl groups exhibit distinct orientations. Functional assays reveal the critical role of extracellular loop 3 in A₃AR's ligand selectivity and receptor activation. Key mutations, including His$^{3.37}$, Ser$^{5.42}$, and Ser$^{6.52}$, in a unique sub-pocket of A₃AR, significantly impact receptor activation. Comparative analysis with the inactive A₂AAR structure highlights a conserved receptor activation mechanism. Our findings provide comprehensive insights into the molecular recognition and signaling of A₃AR, paving the way for designing subtype-selective adenosine receptor ligands.

The adenosine receptor subfamily of G protein-coupled receptors consists of four subtypes: A₁, A₂A, A₂B, and A₃[1,2]. These receptors are activated by the endogenous ligand, adenosine, to transduce downstream signals that mediate a number of important physiological and pathological functions including immunomodulation, energy balance, cardiac function, and neuroprotection[3–5]. The gene of A₃AR was firstly cloned in 1991[6] and characterized as a subtype of adenosine receptor in 1993[1]. It is expressed in various tissues including the brain, heart, lungs, liver, kidneys, and immune cells[7]. A₃AR participates in regulating cardiac function, vasodilation, inhibition of inflammation, protection against ischemia-reperfusion injury, and suppression of oxidative stress. Additionally, A₃AR is highly expressed in several tumor types, making it as a promising therapeutic target for suppressing cancer cell proliferation[7–9].

A₁AR and A₃AR preferentially couple to the inhibitory G protein (Gᵢ), leading to the suppression of adenylate cyclase activity and a reduction in intracellular cyclic AMP levels, contrasting with the stimulatory G protein (Gₛ) signaling triggered by A₂AAR and A₂BAR activation[2]. The structure of adenosine has inspired the design of various agonists and antagonists targeting A₃AR, particularly for cancer, inflammation, and pain management[10]. Studies highlight that alterations at the N⁶ position of the purine ring and the 5'-N position of the ribose group enhance the potency and selectivity of A₃AR agonists[11–13]. Notably, N⁶-methyladenosine (m⁶A), a methylated adenosine metabolite, emerged as a potent A₃AR agonist[14]. CF101 and CF102 are representatives of such modification strategy with similar nucleoside core structure and only one chloro-substituent difference, both demonstrate high affinity and selectivity for A₃AR[15–17].

¹State Key Laboratory of Drug Research, Shanghai Institute of Materia Medica, Chinese Academy of Sciences, Shanghai, China. ²University of Chinese Academy of Sciences, Beijing, China. ³Lingang Laboratory, Shanghai, China. ⁴School of Pharmaceutical Science and Technology, Hangzhou Institute for Advanced Study, University of Chinese Academy of Sciences, Hangzhou, China. ⁵School of Life Science and Technology, ShanghaiTech University, Shanghai, China. ⁶Shandong Laboratory of Yantai Drug Discovery, Bohai Rim Advanced Research, Institute for Drug Discovery, Yantai, China. ⁷These authors contributed equally: Hongmin Cai, Shimeng Guo, Youwei Xu, Jun Sun. ✉e-mail: caihongmin@simm.ac.cn; xxie@simm.ac.cn; eric.xu@simm.ac.cn

These effective orally compounds have shown promise in disrupting key signaling pathways in cancer and inflammatory cells[10]. CF101 has demonstrated efficacy in Phase III trials for psoriasis and rheumatoid arthritis, while CF102 is being evaluated for hepatocellular carcinoma and non-alcoholic steatohepatitis[18,19]. The broad expression of adenosine receptors poses challenges in designing subtype-selective compounds[20,21]. The lack of structural information for $A_3AR$, unlike other adenosine receptor subtypes, limits our understanding of its specific signaling mechanisms and impedes structure-based drug design.

Here, we present the cryo-EM structures of $A_3AR$ bound to the $G_i$ protein in the presence of CF101 and CF102. These structures reveal the mechanisms of ligand recognition and activation in $A_3AR$, providing valuable insights for designing effective, targeted therapies for conditions like cancer and inflammation.

## Results and discussion
### Overall structures of the complexes
CF101 and CF102 are $A_3AR$ agonists that contain modifications to the ribose and adenine moieties, which confer their potent binding to $A_3AR$. Specifically, CF101 and CF102 have a 5′-N-methylcarboxamide substitution on the ribose group and a $N^6$-(3-iodobenzyl) substitution on the adenine base (Fig. 1a). These combined modifications result in significantly higher $A_3AR$ potency compared to the endogenous $A_3AR$ agonist adenosine. To ensure the specificity of our experiments in the context of HEK293 cells, which are known to express high levels of $A_1AR$, $A_{2A}AR$, and $A_{2B}AR$ but not $A_3AR$, we employed NanoBiT association assays. These assays were crucial in determining the selectivity of CF101 and CF102 for $A_3AR$, as opposed to other adenosine receptor subtypes (Fig. 1b–d). While adenosine activated four subtypes with similar micromolar potencies, CF101 and CF102 displayed strong

potency of ~3 nM on $A_3AR$ but had weak or negligible response on other subtypes of adenosine receptors.

We used HiBiT tether approach to stabilize the full-length $A_3AR$-G protein complexes, as it has been used for many GPCR structural studies[22–24] (Supplementary Fig. S1). The large NanoLuc domain (LgBiT) and small high affinity fragment (HiBiT) was fused at the C-terminal of $A_3AR$ and Gβ, respectively. Meanwhile, $A_3AR$ used in this study had an N-terminal thermostabilized apocytochrome b562RIL (BRIL) fusion to enhance its expression, which is co-expressed with G protein subunits and scFv16, an antibody fragment that is used to further stabilize the receptor G protein complex. For the CF101-$A_3AR$-$G_i$ complex, data from 20,779 movies comprising 271,323 particles was used to determine the structure at 3.29 Å resolution (Supplementary Fig. S2, Supplementary Table S1). For the CF102-$A_3AR$-$G_i$ complex, data from 13,581 movies yielding 283,561 particles was used to determine the structure at a resolution of 3.19 Å (Supplementary Fig. S3, Supplementary Table S1). The structures of the CF101/CF102-$A_3AR$-$G_i$ complexes revealed that the ligands occupy the orthosteric binding pocket, with the core structures modeled clearly into the cryo-EM density at the center of the receptor transmembrane helical domain (TMD) (Fig. 1e–h).

The structures showed the canonical seven-transmembrane architecture for $A_3AR$, with the intracellular domains occupied by the α5 helix of $Gα_i$ for $G_i$ coupling. The density maps enabled modeling of most of the structures, except for $A_3AR$ N-terminus residues M1-L8, third intracellular loop N211-Y222, C-terminus V301-E318, and the alpha-helical domain of $Gα_i$. The extracellular loop M151-S165 was also less defined but the backbone could be established (Supplementary Fig. S4). Aside from these regions, the models were well-resolved. Overall, the two agonist-bound complexes were highly similar, with 0.593 Å root mean square deviation (RMSD) for the whole receptor.

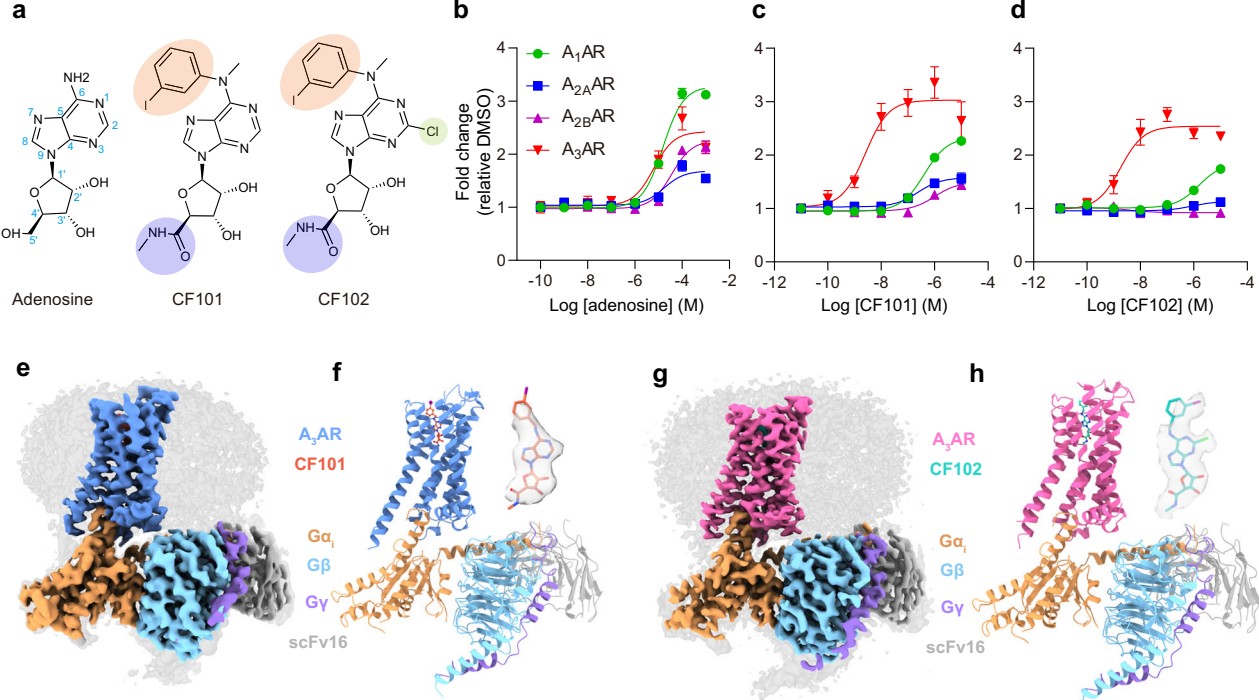

**Fig. 1 | Cryo-EM structures of CF101-$A_3AR$-$G_i$ and CF102-$A_3AR$-$G_i$ complexes.** **a** Chemical structures of the adenosine, CF101 and CF102 are provided, highlighting modifications at the 5′-N-methylcarboxamide in the ribose group, as well as the $N^6$ and C2 positions of the adenosine group. The atom numbering is indicated in blue. CF101, is also named IB-MECA and $N^6$-(3-iodobenzyl)adenosine-5′-N-methyluronamide. CF102, is also named Cl-IB-MECA and 2-chloro-$N^6$-(3-iodobenzyl)adenosine-5′-N-methyluronamide. NanoBiT association assays monitoring ligand activity on adenosine receptors for adenosine (**b**), CF101 (**c**) and CF102 (**d**), respectively. Data shown are mean ± S.E.M. of three independent experiments (n = 3). Source data are provided as a Source Data file. Cryo-EM map (**e**) and model (**f**) of the CF101-$A_3AR$-$G_i$ complex, with inset showing CF101 density. The density map in the inset is shown at 0.232 threshold. Cryo-EM map (**g**) and model (**h**) of the CF102-$A_3AR$-$G_i$ complex, with inset showing CF102 density. The density map in the inset is shown at 0.17 threshold. Subunits are colored as indicated.

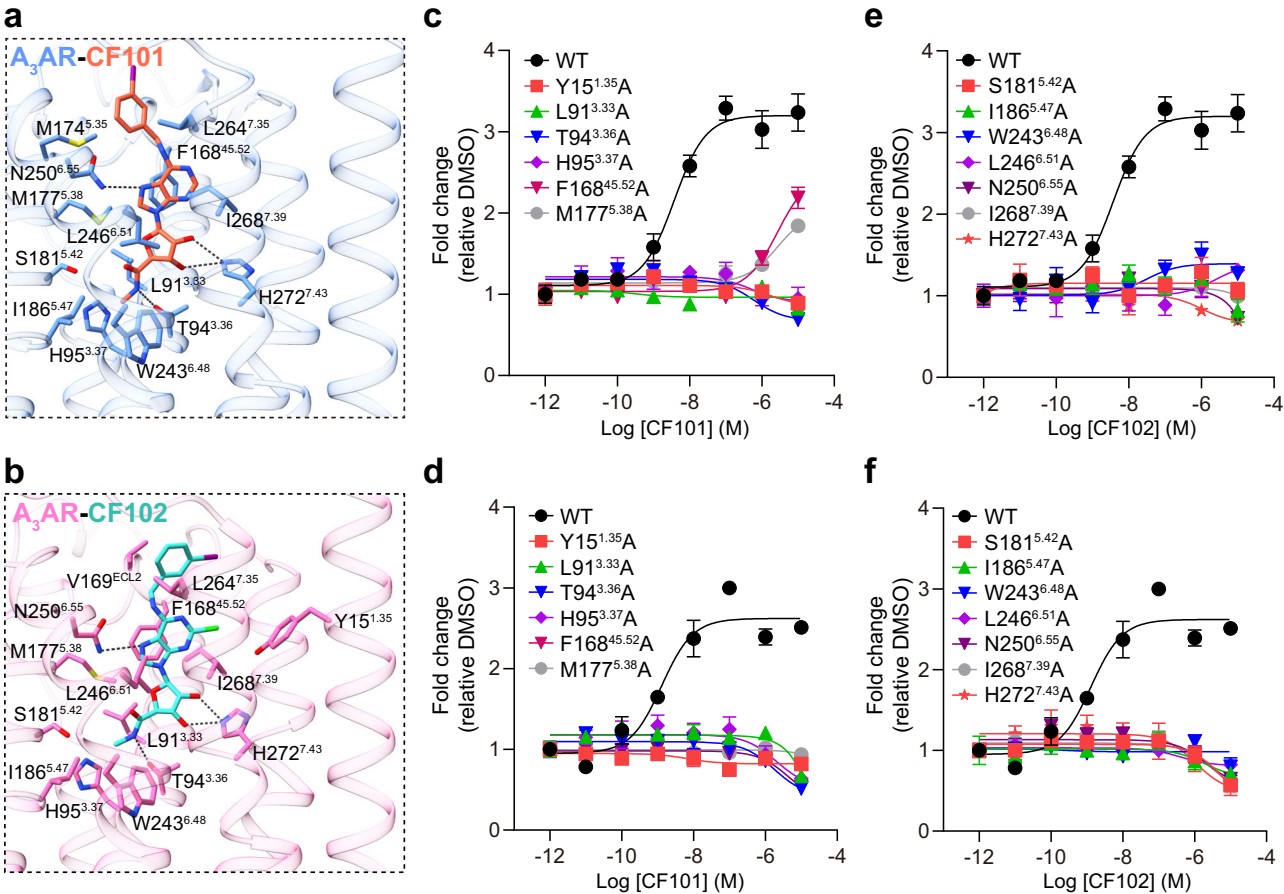

**Fig. 2 | The orthosteric binding pocket.** Detailed interactions between A$_3$AR and CF101 (**a**) or CF102 (**b**) from the membrane plane. Residues involved in ligand interaction are colored blue and pink in two complexes, respectively. Black dashed lines indicate hydrogen bonds. Dose-response curves of mutants of A$_3$AR induced by CF101 (upper panels, **c**, **e**) or CF102 (lower panels, **d**, **f**) using NanoBiT assay. Data shown are mean ± S.E.M. of three independent experiments (*n* = 3). Source data are provided as a Source Data file.

## Binding mode of CF101/CF102 in A$_3$AR orthosteric site

The A$_3$AR agonists CF101 and CF102 bind at conserved orthosteric pocket forms by ECL2, TM3, TM5, TM6 and TM7, akin to the endogenous ligand adenosine bound to other adenosine receptor subtypes (Fig. 2a, b). However, the orientations of the modified 3-iodobenzyl moieties differ between CF101 and CF102. The adenine core mediates conserved receptor interactions commonly seen in other adenosine receptors[23,25,26]. Notably, the adenine pyrimidine forms π-stacks against F[45.52], and the F[45.52]A mutation greatly affected the ability of CF101/CF102 to induce the receptor activation in the NanoBiT association assay (Fig. 2c–f, Supplementary Table S2). Additionally, 2' and 3' hydroxyl group in ribose and purine group form hydrogen bonds with polar side chains at positions 3.36, 6.55 and 7.43, which are conserved and critical for recognition of nucleoside ligands by adenosine receptors (Fig. 2c–f, Supplementary Table S2).

The ligand binding pocket is mainly comprised of hydrophobic residues, including position 3.33, 5.38, 5.47, 6.48, 6.51 and 7.39, which form hydrophobic contacts that are important for CF101/CF102 potencies (Fig. 2c–f, Supplementary Table S2). Alanine mutations at these positions severely reduced agonists' ability to induce receptor activation. His[3.37] and Ser[5.42] participate van der Waals contacts with the bound ligands, their alanine mutations also affected activity (Fig. 2c–f, Supplementary Table S2). To confirm the functional data, cAMP accumulation assays were carried out to assess the agonist activity (Supplementary Fig. S5, Supplementary Table S3). The results from the NanoBiT association assay and cAMP accumulation assay were consistent. The side chains from M174[5.35] and L264[7.35] in the receptor form

hydrophobic interactions with the 3-iodobenzyl group extended from the N[6] position of the adenosine base of CF101. In contrast, the corresponding group of CF102 is surrounded by V169[ECL2] and L264[7.35] from the receptor. Alanine mutations on these residues did not significantly affect the potency of the compounds on A$_3$AR (Supplementary Fig. S6, Supplementary Table S2), suggesting that the 3-iodobenzyl substituents may exist alternative states at the receptor extracellular domains. This demonstrates that the N[6] position may accommodate various substituted groups through distinct conformations in the A$_3$AR pocket.

CF102 is a 2-chloro derivative of CF101. In CF102-bound A$_3$AR, Y15[1.35] is situated near the 2-chloro group of CF102 (Supplementary Fig. S7a). The Y15[1.35]A mutation in A$_3$AR abolished the agonist activity of both CF101 and CF102 (Fig. 2c, d). However, the Y15[1.35]F mutant only slightly impacted the potency of CF101 and CF102 (Supplementary Fig. S7b, c). Y15[1.35] forms extensive π-π contact with Y265[7.36] in TM7. The Y265[7.36]A mutant also affected the receptor's ability to bind CF101 or CF102 (Supplementary Fig. S7). This implies that Y15[1.35] likely plays a critical role in maintaining the stability and structural integrity of A$_3$AR, thus affecting both CF101 and CF102 binding to the receptor. Additionally, modifications at the 2-position of adenosine tend to be well tolerated for A$_3$AR binding[16], whether incorporating a small or large substituent, or even linking it to the N[6] moiety to form a macrocycle[27]. Elucidation of these subtle ligand and receptor interaction variations thus provides molecular insight into the conformational adaptability and binding poses governing molecular recognition by A$_3$AR.

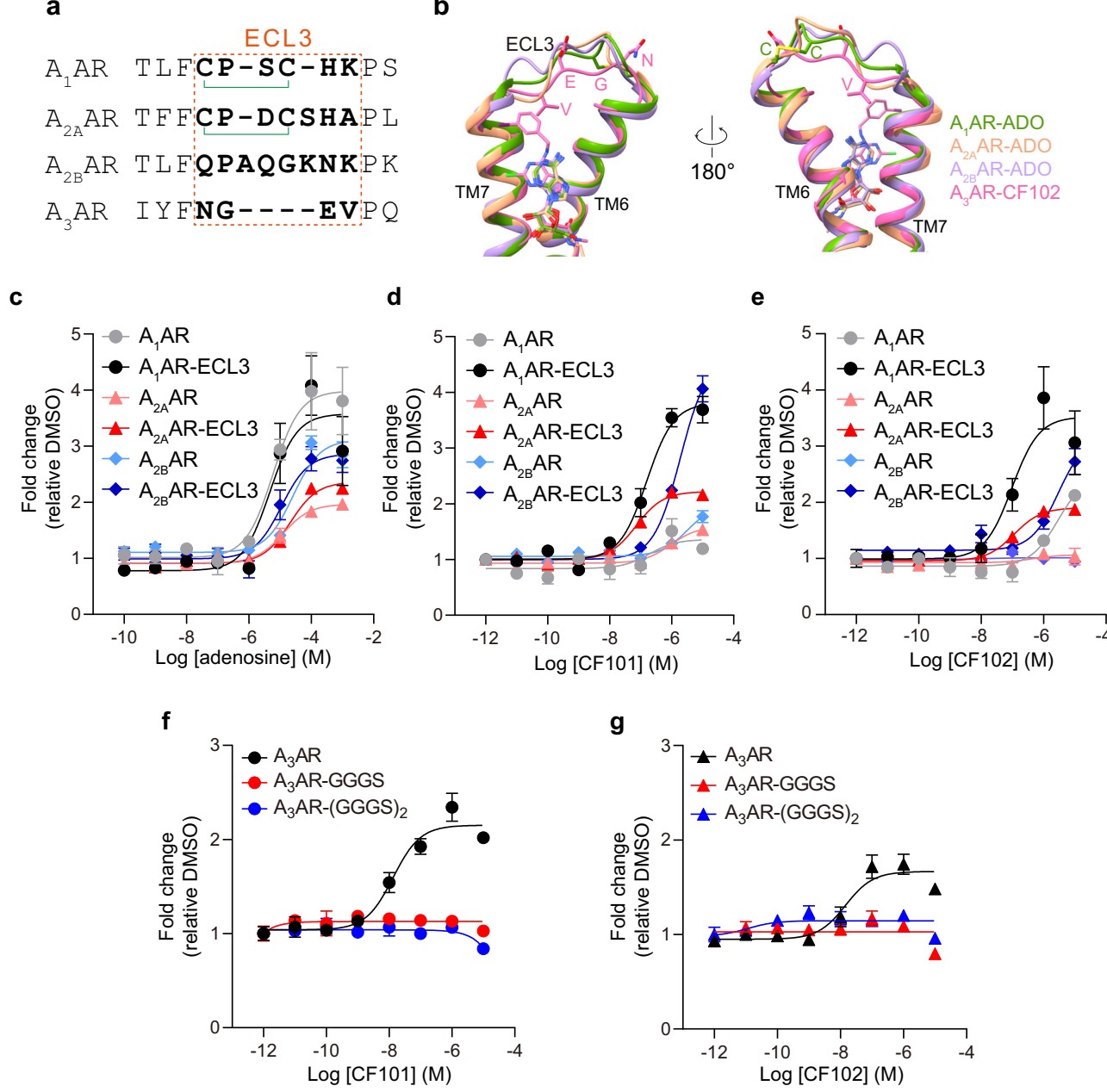

**Fig. 3 | Swapping ECL3 on adenosine receptor subtypes. a** sequence alignment of ECL3 among adenosine receptors. The disulfide bond was shown as green linker. **b** Superimposed structures of adenosine receptors reveal that $A_3AR$ has the shortest ECL3. The residues in $A_3AR$ are shown in pink. The residues formed disulfide bond on ECL3 in $A_1AR$ were shown in green. Other TMs were omitted. **c–e** Assessing the effects of adenosine, CF101, and CF102 on $A_1AR$, $A_{2A}AR$, and $A_{2B}AR$, along with their corresponding mutants containing the swapped ECL3 from the $A_3AR$ using NanoBiT assays. The results were from three independent experiments. Data shown are mean ± S.E.M. of three independent experiments ($n = 3$). Source data are provided as a Source Data file. **f, g** Assessing the effects of CF101 and CF102 on $A_3AR$ and its mutants with flexible ECL3 using NanoBiT assays. Data shown are mean ± S.E.M. of three independent experiments ($n = 3$). Source data are provided as a Source Data file.

## The role of ECL3 in $A_3AR$ subtype selectivity

CF101 and CF102 show high selectivity on $A_3AR$ rather than other subtypes (Fig. 1c, d). Analysis the sequence of adenosine receptors reveals strong conservation within TMs, while the extracellular loops diverge among subtypes (Supplementary Fig. S8). ECL1 shows relatively distant from the orthosteric site. The residue F168[45.52] in ECL2 of adenosine receptors provides critical π-π interactions with both agonists and antagonists binding to these receptors. However, $A_3AR$ possesses a shorter ECL3 than other subtypes (Fig. 3a). The shorter ECL3 may rigidify $A_3AR$ to minimize its conformational changes for ligand binding (Fig. 3b).

To assess the role of ECL3 in $A_3AR$, we engineered chimeric receptors by grafting ECL3 from $A_3AR$ onto the backbones of other adenosine receptors. NanoBiT assays were performed to test the binding abilities of adenosine, CF101, and CF102 to wide-type or chimeric adenosine receptors (Fig. 3c–e). The result showed that the three chimeric adenosine receptors did not show increased binding ability to the endogenous ligand adenosine (Fig. 3c, Supplementary Table S4). However, the three ECL3-chimeric receptors gained the ability to bind CF101 and CF102 with increased efficacy or potency (Fig. 3d, e, Supplementary Table S3). These findings suggest that ECL3 could serve as a structural factor mediating the selective recognition

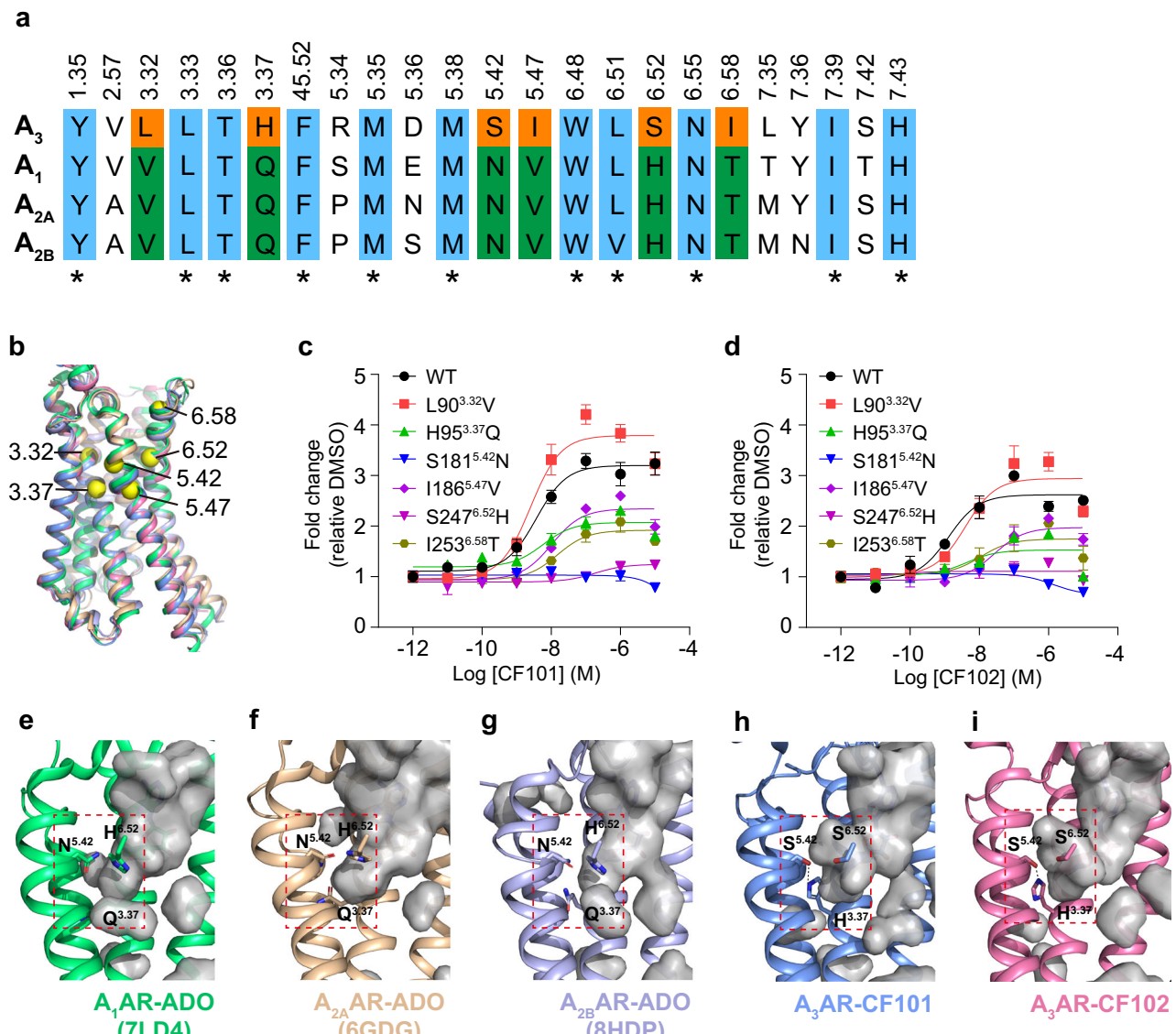

**Fig. 4 | Key residues in the A₃AR binding pocket. a** Aligning the residues in the orthosteric binding pocket among the adenosine receptors. The conserved residues were colored in blue, and stars were used as markers. The unique residues in A₃AR, distinct from other adenosine receptors subtypes were colored in orange, while residues in corresponding positions in other subtypes were colored in green. All residues were annotated based on GPCR Ballesteros-Weinstein numbering scheme. **b** In the superposition of adenosine receptors, the unique residues in A₃AR, distinct from those in other adenosine receptors, are represented as yellow spheres. **c**, **d** Effects of CF101/CF102 on A₃AR mutants containing swapped residues

from other adenosine receptors by NanoBiT assay. Data shown are mean ± S.E.M. of three independent experiments ($n = 3$). Source data are provided as a Source Data file. **e**–**i** The binding cavities of the adenosine receptors were generated in PyMOL and depicted in gray. In A₃AR, a subpocket is formed by His[3.37], Ser[5.42], and Ser[6.52], while these positions are conserved as Gln[3.37], Asn[5.42], and His[6.52] in other adenosine receptor subtypes (His, H; Ser, S; Gln, Q; Asn, N). In **h** and **i**, dashed lines depict the hydrogen bonds between His[3.37] and Ser[5.42]. The names of the receptors and their associated PDB codes[23,26,45] are indicated below each model.

CF101 and CF102 by A₃AR. The significance of ECL3's length and amino acid composition in A₃AR's ligand binding was further investigated through mutations. We mutated the original four ECL3 residues of A₃AR to GGGS or (GGGS)₂ that has the same length as ECL3 of A₂BAR. Neither mutant above showed any binding ability to CF101 and CF102 (Fig. 3f, g), suggesting that both the specific length and the unique amino acid sequence of ECL3 play critical roles in the selective binding of ligands to A₃AR, underscoring the complexity of ligand-receptor interactions in this context.

The proximity of the ECL3 to the N⁶ position in adenosine is likely a crucial factor in the selectivity of A₃AR for N⁶-modified adenosine derivatives, as indicated by structure-activity relationship (SAR) studies[15,16]. Substituents at the N⁶ position, whether too small or overly bulky, can adversely affect the potency and affinity of ligands for A₃AR.

This relationship underscores the importance of ECL3 in ligand recognition, as the N⁶ position extends into A₃AR's binding pocket near ECL3. Understanding these intricate structural interactions is key for discerning the selectivity mechanisms of structurally similar ligands at different adenosine receptors.

## Residues in binding pocket across adenosine receptors

Among adenosine receptors, A₃AR stands out with the lowest sequence identity compared to other subtypes. This distinction is particularly evident in the orthosteric binding pocket (Fig. 4a), where A₃AR's unique residues at specific positions contribute to its selective ligand binding. Notably, positions 3.32, 3.37, 5.42, 5.47, 6.52, and 6.58 feature different amino acids in A₃AR compared to A₁, A₂A, and A₂B receptors (Fig. 4b, Supplementary Fig. S9). Mutations at these

positions to their counterparts in other subtypes were conducted to evaluate their impact on CF101 and CF102 binding and activity. NanoBiT assays and cAMP accumulation assays were utilized to cross-confirm the effects of the mutations (Fig. 4c, d, Supplementary Fig. S10, Supplementary Tables S2 and 3).

We found that changing the leucine at 3.32 to valine, similar to other subtypes, had no significant effect on the activity of CF101 and CF102 in NanoBiT assay, likely due to their comparable hydrophobic properties (Fig. 4c, d). However, mutations at positions 5.47 and 6.58 altered the receptor activation, indicating the importance of side chain length at these positions for ligand binding (Fig. 4c, d).

Furthermore, the hydrogen bond formation between $H95^{3.37}$ and $S181^{5.42}$ in $A_3AR$, which was absent in other subtypes, appears critical (Fig. 4e–i). Mutations $H95^{3.37}Q$ and $S247^{5.42}N$ significantly impacted CF101 and CF102 activities (Fig. 4c, d), highlighting the importance of these residues in ligand-receptor interaction. The mutation of $S247^{6.52}$ to histidine also reduced ligand activity, suggesting the influence of steric and electronic properties of the side chains (Fig. 4c, d, Supplementary Table S2).

Residues $H95^{3.37}$, $S181^{5.42}$ and $S247^{6.52}$ form a unique sub-pocket in $A_3AR$ to accommodate the 5'-N-methylcarboxamide from the ribose (Fig. 4h, i, Supplementary Fig. S11). The mutational results implicate this sub-pocket might serve as a structural determinant for stabilizing CF101 and CF102 in $A_3AR$ versus other subtypes. Our results above with NanoBiT assay were replicated with traditional cAMP accumulation assays (Fig. 4c, d, Supplementary Fig. S10), further demonstrating that how minor sequence variations in receptors can significantly influence their conformations and ligand binding specificity.

## Activation mechanisms of $A_3AR$

Structural comparisons between active, agonist-bound $A_3AR$ complexes and an inactive, antagonist-bound $A_{2A}AR$ structure (PDB ID: 4EIY)[28] reveal classical hallmarks of conformational changes associated with GPCR activation[29,30]. Notably, the $A_3AR$ structures exhibit an outward movement of TM6 compared to inactive $A_{2A}AR$, shifting 11.6 Å based on measurements of residue $Glu^{6.30}$ at Cα atoms in receptors (Fig. 5a). Additional rearrangements of activation include inward movements of TM1 and TM7 and an upward shift of TM3 in $A_3AR$ relative to inactive $A_{2A}AR$ (Fig. 5b–d).

Detail structural analysis also provide potential mechanism of ligand induced $A_3AR$ activation.

A unique sub-pocket formed by $H^{3.37}$, $S^{5.42}$ and $S^{6.52}$ residues confers selectivity over other adenosine receptor subtypes (Fig. 5e). This facilitates deeper binding of CF101/CF102 compared to $A_{2A}AR$ antagonists, enabling engagement with conserved motifs like the $W^{6.48}$ "toggle switch". Propagation through $P^{5.50}I^{3.40}F^{6.44}$, $D^{3.49}R^{3.50}Y^{3.51}$, and $N^{7.49}P^{7.50}xxY^{7.53}$ motifs transduces rearrangements (Fig. 5f–h), while limited ECL3 flexibility likely assists selective activation. By elucidating the structural transitions from inactive to active $A_3AR$, our findings provide molecular insights connecting specialized agonist recognition to downstream signaling activation.

## G protein coupling of adenosine receptors

Adenosine receptors exhibit differential G protein coupling preferences that correlate with distinct conformational orientations of the associated G proteins[23,25,26]. Structural alignment reveals $A_3AR$-$G_i$ shares better overlay with $A_1AR$-$G_i$ versus $A_{2A}/A_{2B}AR$-$G_s$ (Fig. 6a). The analogous $G_i$-binding modes of $A_3AR$ and $A_1AR$ contrast $A_{2A}/A_{2B}AR$'s $G_s$-coupling preferences, consistent with sequence and functional profiles. Notably, TM6 positioning facilitates differential G protein accommodation, 3.1 Å inward shift enables $A_1/A_3AR$-$G_i$ versus $A_{2A}/A_{2B}AR$-$G_s$ binding (Fig. 6b). Additionally, α5 helix and αN of $G_i$ protein tilt orient differently between complexes, induced by receptors' hydrophobic and polar residue interactions (Fig. 6c, d). The α5 helix of $Gα_s$ subunits in $A_{2A}AR$-$G_s$ displays an 8.6 Å displacement

relative to its orientation in $A_3AR$-$G_i$ complexes based on measurements of the Cα atom of $Gα^{H5.03}$ (Fig. 6c). The αN helix of $Gα_i$ exhibits a 3.3 Å tilt compared to $G_s$ when measuring the Cα of $Gα^{HN.39}$.

Furthermore, different adenosine receptors induced a variation in the N-terminal helix (αN) tilt of the Gα protein (Fig. 6d). The residue at position 34.51 (L/L/L/V, the residue in $A_1$/$A_{2A}$/$A_{2B}$/$A_3$-AR) in receptors is conserved as a hydrophobic residue that forms hydrophobic interactions with the G protein by inserting into the cleft between αN and α5 of the Gα protein (Supplementary Fig. S12a). Besides, sequence alignment of adenosine receptors showed that residues at positions 3.53 (R/A/A/R) and 34.55 (M/G/S/R) revealed different preferences in different subtypes (Fig. 6e, f, Supplementary Fig. S12a). The longer side chains in $G_i$-coupled $A_1AR$ and $A_3AR$ likely triggered more noticeable translocations in the αN and α5 helix to accommodate the $Gα_i$ protein. The main chains from $R^{3.53}$ and $P^{34.50}$ in $A_1AR$ formed a polar interaction with N348 in $Gα_i$ protein (Supplementary Fig. S12b). The side chain of $R^{3.53}$ and $H^{4.39}$ in $A_3AR$ formed a polar interaction with the side chain of N347 and E28 in $Gα_i$ protein, respectively (Supplementary Fig. S12c). Both of the complexes of $A_{2A}AR$/$A_{2B}AR$-$G_s$, H41 in $Gα_s$ formed a polar interaction with the main chain from the receptor's ICL2 (Supplementary Fig. S12d, e). Together, these findings reveal that preferred $G_i$-coupled adenosine receptors adopt conserved $G_i$ protein-binding conformations that differ distinctly from those of $G_s$-coupled adenosine receptor subtypes.

In summary, we have determined cryo-EM structures of the $A_3AR$ bound to selective agonists CF101 and CF102 with heterotrimeric $G_i$ protein. Despite the conserved binding of the core adenosine moiety, the structures revealed differences in the orientations of the $N^6$ substituted groups in CF101 and CF102. We have identified ECL3 and key sub-pocket residues $His^{3.37}$, $Ser^{5.42}$ and $Ser^{6.52}$ that confer selectivity over other adenosine receptor subtypes by structural and mutational studies. Comparison to an inactive $A_{2A}AR$ structure provided insight into the conformational changes associated with $A_3AR$ activation and G protein coupling. By elucidating the molecular mechanisms governing ligand recognition, signaling, and subtype selectivity, the experimentally determined $A_3AR$ structures significantly advance our fundamental understanding of this important drug target. The findings pave the way for structure-guided design of selective ligands targeting adenosine receptors subtypes for the treatment of cancer, inflammation, and other diseases.

## Methods
### Construct design
The full-length gene coding human $A_3AR$ was synthesized (Synbio) and subcloned into pFastBac vector using ClonExpress II one step cloning kit (Vazyme Biotech, C112). A hemagglutinin signal peptide and thermostabilized apocytochrome b562RIL (BRIL) were fused at the N-terminal of $A_3AR$ to enhance receptor expression. To enhance complex stability, a NanoBiT tethering approach was used where an LgBiT domain was fused to the C-terminal of the receptor[22]. A dual maltose-binding protein was linked after LgBiT through a tobacco etch virus protease site (TEV site) for further cleavage. A dominant-negative mutant of bovine $Gα_i$ containing G203A/A326S[31] mutations was generated to stabilize the heterotrimeric $Gα_iβγ$ protein. Rat Gβ1 was fused with a HiBiT at C-terminal for structural complementation of LgBiT to form a NanoBiT. The single-chain variable fragment scFv16 was applied to bind the $Gα_iβγ$ protein for stabilization[32]. $Gα_i$, Gβ1-HiBiT, Gγ, and scFv16, were cloned into pFastBac vector (Supplementary Fig. 1a), respectively.

### Protein expression and purification
The recombinant $A_3AR$, $Gα_i$, Gβ1-HiBiT, Gγ, and scFv16 were co-expressed in *Trichoplusia ni* High Five insect cells using the Bac-to-Bac baculovirus expression system. High Five cells were co-infected with the baculovirus at a cell density of $3.5 \times 10^6$ cells per milliliter.

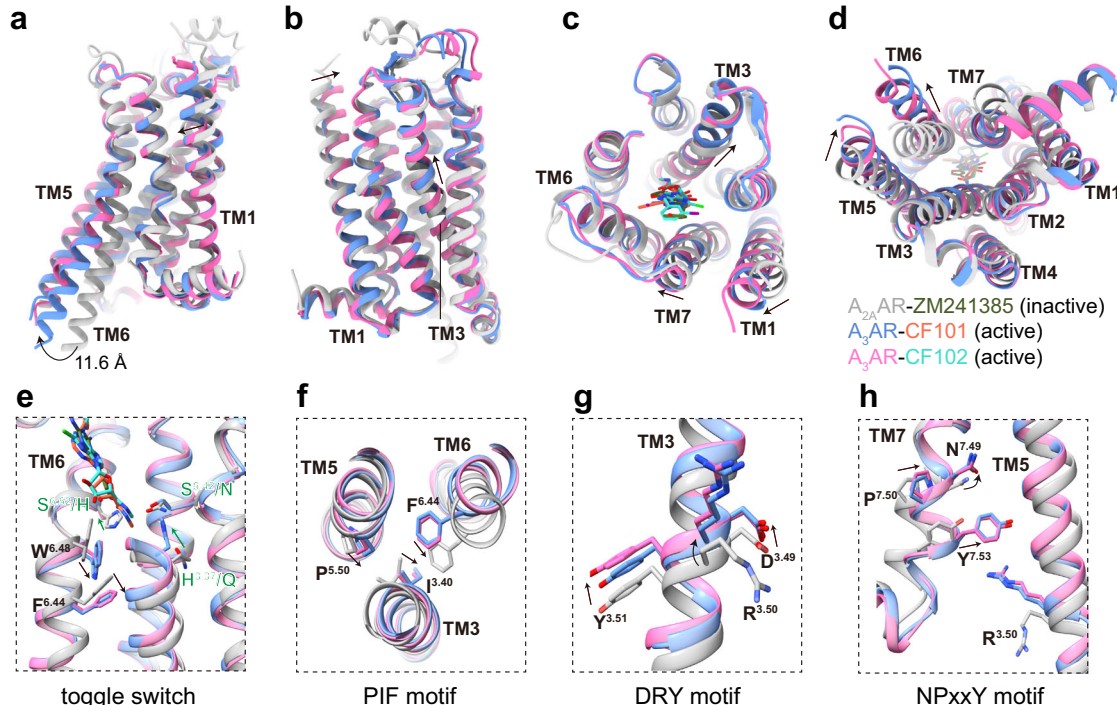

**Fig. 5 | A₃AR activation mechanism. a**, **b** Superposition of active A₃AR-CF101/CF102 complexes (blue/pink) with inactive A₂ₐAR-ZM241385 complex (gray, PDB ID 4EIY). Comparison of extracellular (**c**) and cytoplasmic (**d**) views of active A₃AR and inactive A₂ₐAR. **e**–**h** Conformational changes in conserved motifs, including the toggle switch, PIF, DRY and NPxxY, upon CF101/CF102 binding to A₃AR relative to inactive state of A₂ₐAR-ZM241385. Arrows indicate movement directions. In **e**, The sub-pocket in A₃AR is formed by residues at position 3.37, 6.52 and 6.52. The residues at these positions from both A₃AR and A₂ₐAR were labeled in green.

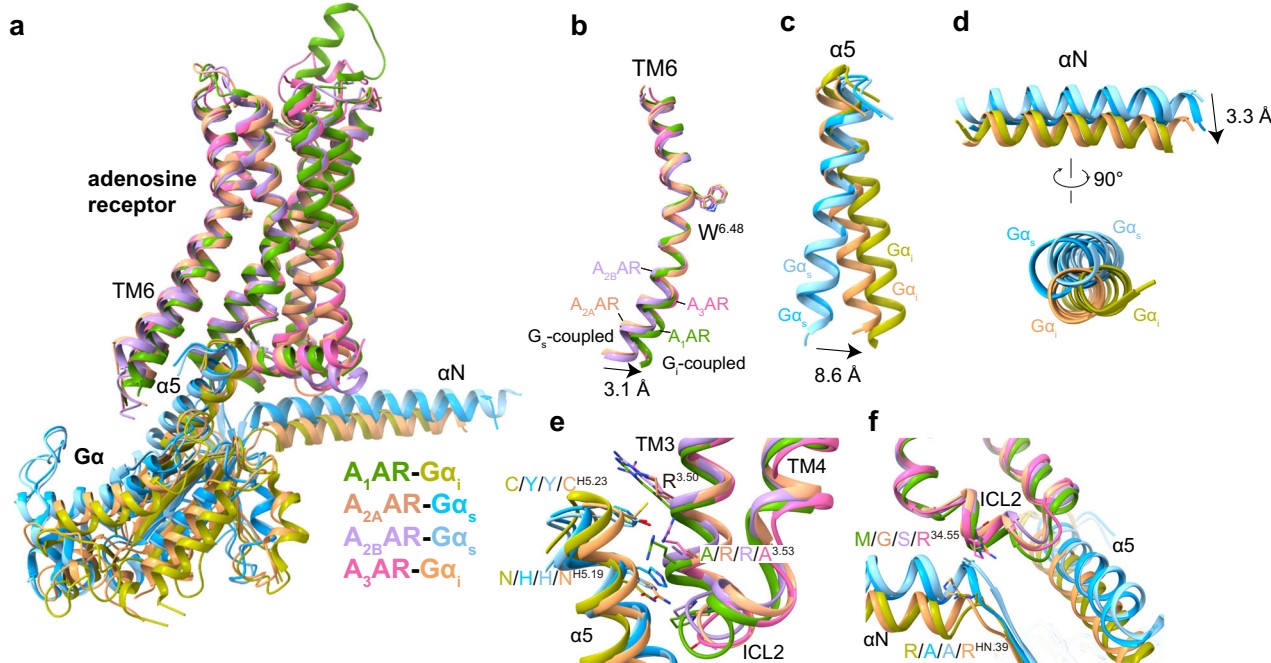

**Fig. 6 | G protein coupling of adenosine receptors. a** Comparison of adenosine receptor and Gα protein conformations in A₁/A₃AR-Gᵢ and A₂ₐ/A₂ᵦAR-Gₛ complexes. Omitting the Gβ and Gγ subunits. The PDB codes for A₁AR, A₂ₐAR and A₂ᵦAR are 7LD4, 6GDG and 8HDP, respectively. **b** Conformational comparison of TM6 in adenosine receptors, with reference to the toggle switch residue W⁶·⁴⁸ in TM6 of receptor. **c**, **d** Conformational comparison of the α5 helix and αN helix in G protein among adenosine receptor-G protein complexes. Arrows indicate movement directions. **e**, **f** Distinguishing residues on TM3 and ICL2 in adenosine receptor subtypes that participate in G protein coupling are highlighted. Components of adenosine receptors-G protein complexes are colored as indicated.

Fourty-eight hours later, the infected cells were harvested and stored at −80 °C until used.

For the purification of the CF101-$A_3AR$-$G_i$ complex, cells pellets were thawed and resuspended in Buffer A (100 mM NaCl, 20 mM HEPES, pH 7.5) supplemented with protease inhibitor cocktail (TargetMol, C0001). Cells were lysed by dounce homogenization (Sigma-Aldrich, D9188) followed by centrifugation to remove unsoluble materials. The pellets were resuspended in Buffer B (100 mM NaCl, 10 %(v/v) glycerol, 20 mM HEPES, pH 7.5) supplemented with 10 mM $MgCl_2$, 5 mM $CaCl_2$, 0.2 mM Tris-(2-carboxyethyl)phosphine (TCEP, Hampton Research, HR2-801) and protease inhibitor cocktail. We formed the complexes by rotating the samples at room temperature for 1 h after addition of 25 mUnit/mL apyrase and 10 µM CF101 (MedChemExpress, HY-13591). After incubation, the sample was solublized in 0.5 %(w/v) lauryl maltose neopentylglycol (LMNG, anatrace, NG310) and 0.1%(w/v) cholesteryl hemisucinate (CHS, anatrace, CH210) for 3 h at 4 °C. The supernatant was clarified by centrifugation at 100,000× g for 40 min. The supernatant was incubated with dextrin beads 6FF (Smart-Lifesciences, SA02601L) for 3 h at 4 °C. The beads were loaded onto a gravity column and washed with 20 column volumes of Buffer C (100 mM NaCl, 2 mM $MgCl_2$, 10 µM CF101, 0.2 mM TCEP, 0.01 %(w/v) LMNG, 0.002 %(w/v) CHS, 20 mM HEPES, pH 7.5). The complex was eluted with Buffer C supplemented with 10 mM maltose and further concentrated using 100 kDa molecular weight cut-off concentrator. TEV protease was added to the concentrated protein at 4 °C overnight to cleave dual maltose binding protein from fusion protein. After digestion, sample was loaded onto Superdex 200 Increase 10/300 GL column (Cytiva, 28-9909-44) with Buffer D (100 mM NaCl, 2 mM $MgCl_2$, 10 µM CF101, 0.1 mM TCEP, 0.00075 %(w/v) LMNG, 0.00025 %(w/v) glyco-diosgenin, 0.0002 %(w/v) CHS, 20 mM HEPES, pH 7.5). The desired fractions were pooled and concentrated to 5–8 mg/mL for cryo-EM sample preparation. The purification procedures of CF102-$A_3AR$-$G_i$ complex were almost the same as in CF102-$A_3AR$-$G_i$ complex preparation, while the CF101 compounds was replaced by CF102 (TargetMol, T6884).

## Cryo-EM data collection
Cryo-EM grids were prepared with the Vitrobot Mark IV plunger (FEI) set to 8 °C and 100% humidity. Three-microliters of the CF101-$A_3AR$-$G_i$ complex were applied to glow- discharged Quantifoil R1.2/1.3 holey carbon grids. The sample was incubated for 10 s on the grids before blotting for 3.5 s (double-sided, blot force 1) and flash-frozen in liquid ethane immediately. The same conditions were used for the CF102-$A_3AR$-$G_i$ complex sample.

For CF101-$A_3AR$-$G_i$ complex, three datasets comprising 20,779 movies were collected on a Titan Krios equipped with a Gatan K3 direct electron detection device at 300 kV with a magnification of 105,000 corresponding to a pixel size 0.824 Å. Image acquisition was performed with EPU Software (FEI Eindhoven, Netherlands). We collected a total of 36 frames accumulating to a total dose of 50 e⁻ Å⁻² over 2.5-s exposure.

For CF102-$A_3AR$-$G_i$ complex dataset, two datasets totaling 13,581 movies were collected on a Titan Krios equipped with a Gatan K3 detector at 300 kV with a magnification of 105,000 and pixel size of 0.824 Å, using EPU Software (FEI Eindhoven, Netherlands). Thirty-six frames were collected over a 2.5-s exposure to a dose of 50 e⁻ Å⁻².

## Image processing
MotionCor2 was used to perform the frame-based motion-correction algorithm to generate drift-corrected micrograph for further processing, and CTFFIND4 provided estimation of contrast transfer function (CTF) parameters[33,34].

For the CF101-$A_3AR$-$G_i$ dataset, the previously resolved structure of BAY 60-6583-$A_{2B}AR$-$G_s$[23] was used as a reference for automatic particle picking in RELION 3.0[35]. Particle picking and extraction yielded 4,550,294 particles after 2D classification clearance, which were imported into CryoSPARC[36]. Four rounds of 2D classification selected 1,267,837 particles, followed by two rounds of 3D heterogenous refinement giving 982,833 particles. After two additional rounds of 2D classification and two rounds of heterogenous refinement, 271,323 particles were refined to a structure at 3.29 Å global resolution using non-uniform refinement (Supplementary Fig. 2).

For CF102-$A_3AR$-$G_i$ complex dataset, the BAY 60-6583-$A_{2B}AR$-$G_s$ structure[23] was again used for reference-based particle picking. 4,090,959 and 4,833,382 particles were autopicked and extracted from Dataset 1 and Dateset 2, respectively. For Dataset 1, two rounds of 2D classification were used to separate out 1,070,085 particles. Masked 3D classification on the receptor part was used to separate out 175,747 particles that resulted to a clearer density of $A_3AR$. For Dataset 2, two rounds of 2D classification and two rounds of 3D classification were performed to separate out 246,392 particles. After clearance, the remained particles from two datasets were combined and subjected to alignment-free 3D classification. 283,561 particles were remained and transferred in CryoSPARC[36]. One round of heterogenous refinement yielded a final 102,581 particles were refined to a structure at 3.19 Å global resolution using non-uniform refinement (Supplementary Fig. 3).

## Model building
An $A_3AR$ structure predicted by AlphaFold2 was used as the starting reference models for receptors building[37]. Structures of $G\alpha_i$, $G\beta$, $G\gamma$ and the scFv16 were derived from PDB entry 7EZH[38] were rigid body fit into the density. All models were fitted into the EM density map using UCSF Chimera[39] followed by iterative rounds of manual adjustment and automated rebuilding in COOT[40] and PHENIX[41], respectively. The model was finalized by rebuilding in ISOLDE[42] followed by refinement in PHENIX with torsion-angle restraints to the input model. The final model statistics were validated using Comprehensive validation (cryo-EM) in PHENIX and provided in the supplementary information, Supplementary Table 1. All structural figures were prepared using Chimera[39], Chimera X[43], and PyMOL (Schrödinger, LLC.).

## NanoBiT assay
To monitor G protein interaction with $A_1AR$, $A_{2A}AR$, $A_{2B}AR$ or $A_3AR$ upon agonist stimulation, a NanoLuc-based NanoBiT enzyme complementation assay was used as previously described[44]. The C terminus of $A_1AR$, $A_{2A}AR$ or $A_{2B}AR$ was fused to SmBiT, while LgBiT was fused to the N terminus of miniG proteins. The C terminus of $A_3AR$ was fused with LgBiT, and the SmBiT was fused to the N terminus of miniG proteins. HEK293 cells were seeded at $4 \times 10^4$ cells/well on 96-well plates and co-transfected with adenosine receptor-SmBiT and LgBiT-G protein plasmid. After 24 h, cells were replaced with 40 µL fresh culture medium without fetal bovine serum. Ten microliter Nano-Glo Live Cell reagent was added followed the manufacturer's protocol (Promega, N2011), and incubated at 37 °C, 5 % $CO_2$ for 5 min. Another 25 µL culture medium containing various concentrations of compounds were added and incubated at room temperature for 10 min before measuring bioluminescence using EnVision multiplate reader (PerkinElmer).

## cAMP accumulation assays
HEK293 cells expressing wide-type (WT) or mutant $A_3AR$ were harvested and resuspended in DMEM containing 500 µM 3-isobutyl-1-methylxanthine (IBMX) at a density of $2 \times 10^5$ cells/ mL. Cells were then plated onto 384-well assay plates at 1000 cells/ 5 µL/ well. Another 5 µL buffer containing 1 µM Forskolin and various concentrations of test compounds were added to the cells. After incubation at room temperature for 15 min, intracellular cAMP level was tested by a LANCE

Ultra cAMP kit (PerkinElmer, TRF0264) and EnVision multiplate reader according to the manufacturer's instructions.

## Cell-surface expression assay

The same constructs were used in the cell-surface expression assays, NanoBiT assays, and cAMP measurements. A human influenza hemagglutinin tag (HA-tag) was fused to the N-terminus of the adenosine receptor and mutant gene sequences in the pcDNA3.0 vector constructs used across the various assays. HEK293 cells were transfected with wild type (WT) or adenosine receptor mutants and then were seeded at $4 \times 10^4$ cells/well on 96-well plates. After 24 h, cells were washed with PBS buffer, fixed with 4 %(w/v) paraformaldehyde for 15 min, and blocked with 2 %(w/v) bovine serum albumin (BSA) for 1 h. Next, cells were incubated with the polyclonal anti-HA antibody (diluted at a ratio of 1:1,000, Sigma-Aldrich, H6908) overnight at 4 °C, followed by 1 h with horseradish peroxidase (HRP)-conjugated anti-rabbit antibody (diluted at a ratio of 1:10,000, Cell Signaling, 7074S) at room temperature. After washing, 50 µL tetramethylbenzidine (Sigma, T0440) was added for 30 min before stopping the reaction with 25 µL 3,3,5,5 - tetramethylbenzidine (TMB) substrate stop solution (Beyotime, P0215). Absorbance at 450 nm was measured on a FlexStation III microplate reader (Molecular Devices).

## Statistical analysis

All functional study data were analyzed in Prism 8 (GraphPad) and presented as means ± S.E.M. from at least three independent experiments. Concentration-response curves were evaluated with a three-parameter logistic equation. $pEC_{50}$ values were calculated using the sigmoid three-parameter equation. Significance was determined by one-way ANOVA followed by multiple comparisons test, and $*P < 0.05$ *vs*. wild-type (WT) was considered statistically significant.

## Reporting summary

Further information on research design is available in the Nature Portfolio Reporting Summary linked to this article.

# Data availability

The data that support this study are available from the corresponding authors upon request. The cryo-EM maps have been deposited in the Electron Microscopy Data Bank (EMDB) under accession codes EMD-37985 ($A_3AR$-CF101-$G_i$ complex) and EMD-37986 ($A_3AR$-CF102-$G_i$ complex). The atomic coordinates have been deposited in the Protein Data Bank (PDB) under accession codes 8X16 [https://doi.org/10.2210/pdb8X16/pdb] ($A_3AR$-CF101-$G_i$ complex) and 8X17 [https://doi.org/10.2210/pdb8X17/pdb] ($A_3AR$-CF102-$G_i$ complex). Previously published structures can be accessed via accession codes 7LD4, 6GDG, 8HDP and 4EIY. Source data are provided with this paper.

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

## Acknowledgements

We thank Wen Hu, Kai Wu and Qingning Yuan from the Advanced Center for Electron Microscopy (Shanghai Institute of Materia Medica, Chinese Academy of Sciences) for their technical supporting and assistance with cryo-EM dataset collection. This project was partially supported by the CAS Strategic Priority Research Program (XDB37030103 to H.E.X.); the National Natural Science Foundation of China (32301004 to H.C., 82121005 to X.X., Y.J., and H.E.X., 32130022 to H.E.X., 82330113 to X.X., 82304579 to S.G., and 32171187 to Y.J.); Shanghai Municipal Science and Technology Major Project (2019SHZDZX02 to H.E.X.); Shanghai Municipal Science and Technology Major Project (H.E.X.); the Lingang Laboratory (LG-GG-202204-01 to H.E.X.); the National Key R&D Program of China (2022YFC2703105 to H.E.X.); the China Postdoctoral Science Foundation (2021M703341, 2023T160662 to H.C.), the Shanghai Postdoctoral Excellence Program (2021423, H.C.).

## Author contributions

H.E.X., X.X. and H.C. conceived the study. H.C. designed the expression constructs, purified the protein complexes. Y.X. and H.C. prepared the grids and performed the cryo-EM data processing and model building with the help from J.L. H.E.X., H.C. and Y.J. analyzed the structures. S.G. J.S. and Z.X. performed the functional studies under the supervision of X.X. H.C. prepared the figures and manuscript. Y.X. and S.G. contributed to manuscript preparation. H.E.X. and H.C. wrote the manuscript with input from all authors. The authors utilized Claude.ai and ChatGPT to assist with correcting grammatical errors.

## Competing interests

H.E.X. is the founder and chairman of the board of Cascade Pharmaceutics. The remaining authors declare no competing interests.
