## [Peer Review File · Nature Communications]

Cryo-EM structures of adenosine receptor A3AR bound to selective agonistsReviewer #1 (Remarks to the Author):

Cai et al. have successfully detailed two cryo-EM structures of the A3AR-Gi complexes, each bound with different adenosine analogues. It is noteworthy that A3AR, among the four adenosine receptor subtypes, was the final one to have its structure determined. The authors highlight the distinct structural features of A3AR, particularly the ligand recognition by ECL3. The structure of the ligand pocket provides a clear explanation for the characteristics of previously developed A3AR-selective agonists, especially in terms of the N6 modification. While this study offers valuable insights for the rational development of potent, A3AR-selective drugs, there are several issues that need addressing before the manuscript can be fully endorsed for publication.

Major:

(1) Functional assay

The functional assay utilized in this study, the NanoBiT association, is inadequately described in the methods section. Essential details such as the construction of the LgBiT-G protein (whether it's derived from the full-length G α subunit or miniG, the position of LgBiT fusion, etc.) are missing. If this is the first report of this assay, it is crucial to include detailed information, including the sequences of the constructs.

Furthermore, the assay may only partially measure G-protein activation, capturing a part of the process ($L + R + G\text{-GDP} \Rightarrow RL + G\text{-GDP} \Rightarrow RLG\text{-GDP} \Rightarrow RLG\text{-GTP} \Rightarrow RL + G\text{-GTP}$). If the authors used a miniG-derived construct, it raises additional doubts about accurately measuring signaling-competent G-protein activation. There is a phenomenon known as 'unproductive coupling,' where the G-protein binds to the ligand-activated receptor but fails to undergo nucleotide exchange (PMID 32817560). Given the situation, it is strongly recommended that the authors conduct additional assays to measure G-protein activation or its downstream signaling, such as BRET-based G-protein dissociation or cAMP measurement, particularly for key mutants.

(2) Parameter analysis

In Supplementary Tables 2 and 3, the presentation of potency values in log scale (pEC₅₀, mean \pm SEM) is preferred over the linear scale (EC₅₀). This is because, in the pharmacology field, log-transformed parameters are known to exhibit a Gaussian distribution. Linear-scale values can be derived from the mean pEC₅₀ if necessary.

Additionally, 'max change' values should be substituted with 'E_{max}' values, calculated as 'Top' minus 'Bottom' from the sigmoidal dose-response curve. The current 'max change' appears to be based on data at a 10 μ M ligand concentration, which could be misleading. Moreover, to ensure accuracy, it is crucial to demonstrate that basal NanoBiT counts remain consistent between the wild-type and the mutant constructs.

(3) Expression analysis

The manuscript would benefit from a comparative analysis of mutant construct expression levels under the same transfection conditions used for the NanoBiT assay. Currently, there is a discrepancy as the NanoBiT assay was conducted under the co-expression of R-SmBiT + G-LgBiT, whereas surface-expression analysis used the HA-R construct alone. Particularly for mutants like V243A, I268A, and H272A, which show lower cell-surface expression, it is crucial to discern whether the attenuated NanoBiT association responses result from expression levels or functional deficits. Therefore, it is strongly recommended to standardize the transfection volumes for the NanoBiT and cell-surface expression assays, including titrations of the WT plasmid, to ensure comparable expression levels.

(4) Few novel knowledge

Figure 5 depicts the structural changes associated with GPCR activation, which are now well-characterized in the literature given the numerous of GPCR structure publications nowadays. To enhance the manuscript's impact, the authors should emphasize the distinctive structural features of A3AR that differentiate it from other GPCRs. In addition, Figure 6 seems to merely conclude that the observed structural differences are attributable to the types of bound G protein. The A3AR structure is

almost perfectly superimposed with that of A1AR. For Figure 6d, it appears that when the G protein regions are superimposed, the purported differences of the α N are not observed.

(5) ECL3 mutant

It will be informative to perform the assay with adenosine.

(6) Y15 interaction

In lines 133-134, the reported 4.4 Å distance between Y15 and the 2-chloro group is noted as a hydrophobic contact. However, this distance may be too great to facilitate effective hydrophobic interactions, especially considering the polar nature of the tyrosine hydroxyl group. To investigate the importance of this interaction further, an experimental mutation of Y15F is recommended. This mutation would disrupt any 'hydrophobic contact' while preserving the aromatic interaction with Y265, which may be the more significant contributor to binding affinity. The anticipated result is that the aromatic interaction predominates over the proposed hydrophobic contact in this context.

Minor:

Introduction/discussion

A reference (PMID 33472058) in which m6A, an adenosine metabolite with the modification at the N6 position (as in CF101 and CF102), is proposed to be an endogenous, A3AR-selective ligand, should be referred in the manuscript.

Throughout the manuscript

If the authors would like to mention the adenosine receptors besides A3AR altogether, use "the other" instead of "other".

Line 84

"NanoBiT tether strategy" should be "HiBiT tether strategy".

Line 119

"comprosed" should be "comprised".

Lines 134-135

No figure was shown for Y265.

Line 156-157

Reference is needed for the statement "According to the reported structure-activity relationship of ligands at the A3AR".

Line 440

"solubliezed" should be "solubilized".

Lines 433-453

Add details for the experimental methods for the CF102-bound A3AR.

Line 460

"CF101" was meant to be "CF102"?

Reviewer #2 (Remarks to the Author):

Cai et.al reported two structures of A3AR in the agonist-binding active state. These are the firstly reported A3 receptor structures, however, the structures of other members (A1AR, A2AAR, A2ABR) in this subfamily have been solved already, including the active state and the inactive state, so it is better to explain the significance of this receptor in detail, compare it with other receptors in this

subfamily, and emphasize the necessity of gaining the A3AR structure, otherwise this work is kind of routine without intriguing scientific insights. Besides, I have two major concerns:

(1) The authors stated that the ECL3 part is very important for ligand recognition. However, in the density map, the ECL3 has no direct interaction with the two agonists CF101 and CF102. It needs more experiments to support this claim. For example, the GGSGGS linker substitution could affect ligand binding or not, or an MD simulation that proves the entry of ligand needs the help of ECL3 of A3AR.

(2) The overall experimental design for validating the mutant effect is only the NanoBit assay. It is better that use multiple assays to cross-confirm the conclusions in this paper. For example, the cAMP assay, BRET assay, SPR, and other classical assays.

REVIEWER COMMENTS

Reviewer #1 (Remarks to the Author):

Cai et al. have successfully detailed two cryo-EM structures of the A3AR-Gi complexes, each bound with different adenosine analogues. It is noteworthy that A3AR, among the four adenosine receptor subtypes, was the final one to have its structure determined. The authors highlight the distinct structural features of A3AR, particularly the ligand recognition by ECL3. The structure of the ligand pocket provides a clear explanation for the characteristics of previously developed A3AR-selective agonists, especially in terms of the N6 modification. While this study offers valuable insights for the rational development of potent, A3AR-selective drugs, there are several issues that need addressing before the manuscript can be fully endorsed for publication.

Response: We sincerely thank Reviewer #1 for the positive feedback and valuable suggestions. Your insights have significantly contributed to enhancing the manuscript's quality. In response to your comments, we have carefully revised the manuscript, addressing each point raised to ensure a more comprehensive and insightful presentation of our findings on the A₃AR-G_i complexes.

Major:

(1) Functional assay

The functional assay utilized in this study, the NanoBiT association, is inadequately described in the methods section. Essential details such as the construction of the LgBiT-G protein (whether it's derived from the full-length G α subunit or miniG, the position of LgBiT fusion, etc.) are missing. If this is the first report of this assay, it is crucial to include detailed information, including the sequences of the constructs.

Furthermore, the assay may only partially measure G-protein activation, capturing a part of the process ($L + R + G\text{-GDP} \Rightarrow RL + G\text{-GDP} \Rightarrow RLG\text{-GDP} \Rightarrow RLG\text{-GTP} \Rightarrow RL + G\text{-GTP}$). If the authors used a miniG-derived construct, it raises additional doubts about accurately measuring signaling-competent G-protein activation. There is a phenomenon known as 'unproductive coupling,' where the G-protein binds to the ligand-activated receptor but fails to undergo nucleotide exchange (PMID 32817560). Given the situation, it is strongly recommended that the authors conduct additional assays to measure G-protein activation or its downstream signaling, such as BRET-based G-protein dissociation or cAMP measurement, particularly for key mutants.

Response: We thank the reviewer for the insightful comments on the NanoBiT assay. We enhance the Methods section by including detailed information on the receptor and G protein construction, including the specifics of the SmBiT and the LgBiT fusion position. The same constructs were used in the NanoBiT assays, cAMP measurements and cell-surface expression assays. Additionally, to address reviewer's concerns about the comprehensive measurement of G-protein activation, we have performed supplementary cAMP measurements for mutants. This provides a more robust validation of G-protein

activation, complementing our NanoBiT assay results.

(2) Parameter analysis

In Supplementary Tables 2 and 3, the presentation of potency values in log scale (pEC₅₀, mean ± SEM) is preferred over the linear scale (EC₅₀). This is because, in the pharmacology field, log-transformed parameters are known to exhibit a Gaussian distribution. Linear-scale values can be derived from the mean pEC₅₀ if necessary.

Additionally, 'max change' values should be substituted with 'E_{max}' values, calculated as 'Top' minus 'Bottom' from the sigmoidal dose-response curve. The current 'max change' appears to be based on data at a 10 μM ligand concentration, which could be misleading. Moreover, to ensure accuracy, it is crucial to demonstrate that basal NanoBiT counts remain consistent between the wild-type and the mutant constructs.

Response: We thank the reviewer for the valuable suggestion regarding the presentation of potency values. We have revised Supplementary Tables 2, 3 and 4 to display potency values in log scale (pEC₅₀, mean ± SEM), aligning with standard practices in pharmacology and ensuring a more accurate interpretation of the data. Additionally, we have replaced 'max change' values with 'E_{max}' values, calculated as 'Top' minus 'Bottom' from the sigmoidal dose-response curve, to provide a clearer understanding of the ligand efficacy in Supplementary Table 4. To address reviewer's concern about consistency in the basal NanoBiT counts, we have verified and ensured that these counts remain consistent between the wild-type and mutant constructs.

(3) Expression analysis

The manuscript would benefit from a comparative analysis of mutant construct expression levels under the same transfection conditions used for the NanoBiT assay. Currently, there is a discrepancy as the NanoBiT assay was conducted under the co-expression of R-SmBiT + G-LgBiT, whereas surface-expression analysis used the HA-R construct alone. Particularly for mutants like V243A, I268A, and H272A, which show lower cell-surface expression, it is crucial to discern whether the attenuated NanoBiT association responses result from expression levels or functional deficits. Therefore, it is strongly recommended to standardize the transfection volumes for the NanoBiT and cell-surface expression assays, including titrations of the WT plasmid, to ensure comparable expression levels.

Response: We thank the reviewer for pointing out the need for a consistent approach in our expression analysis. We have now included a comparative analysis of mutant construct expression levels, ensuring both NanoBiT and cell-surface expression assays are conducted under standardized transfection conditions. We agree with the Reviewer that dramatic expression level change of the receptor may have an impact on NanoBiT association responses. But slight to moderate expression variations show negligible impacts on EC₅₀ and E_{max}, as indicated in the figure below. We adjusted the level of H272A expression, the mutant also greatly affected the ability of CF101/CF102 to induce the receptor activation (**Fig. R1**).

Fig. R1 EC₅₀ values of CF101 on wild-type A₃AR with different expression levels.

(4) Few novel knowledge

Figure 5 depicts the structural changes associated with GPCR activation, which are now well-characterized in the literature given the numerous of GPCR structure publications nowadays. To enhance the manuscript's impact, the authors should emphasize the distinctive structural features of A₃AR that differentiate it from other GPCRs. In addition, Figure 6 seems to merely conclude that the observed structural differences are attributable to the types of bound G protein. The A₃AR structure is almost perfectly superimposed with that of A₁AR. For Figure 6d, it appears that when the G protein regions are superimposed, the purported differences of the αN are not observed.

Response: We appreciate the reviewer's suggestion to emphasize the unique structural features of A₃AR. The revised manuscript now includes a detailed section highlighting the distinct structural aspects of A₃AR that set it apart from other adenosine receptor subtypes, particularly focusing on its activation mechanism. This addition aims to underscore the novel insights our study offers in the context of the existing adenosine receptor structures.

(5) ECL3 mutant

It will be informative to perform the assay with adenosine.

Response: We appreciate the reviewer's recommendation to perform the assay with adenosine and have incorporated this into our revised manuscript. Furthermore, to strengthen our understanding of ECL3's role in A₃AR activity, we have included new experimental data on the glycine-serine replacement in ECL3. This addition provides further evidence supporting the significance of ECL3 in A₃AR's functionality.

(6) Y15 interaction

In lines 133-134, the reported 4.4 Å distance between Y15 and the 2-chloro group is noted as a hydrophobic contact. However, this distance may be too great to facilitate effective hydrophobic interactions, especially considering the polar nature of the tyrosine hydroxyl group. To investigate the importance of this interaction further, an experimental mutation of Y15F is recommended. This mutation would disrupt any 'hydrophobic contact' while preserving the aromatic interaction with Y265, which may be the more significant contributor to binding affinity. The anticipated result is that the aromatic interaction predominates over the proposed hydrophobic contact in this context.

Response: We thank the reviewer for the insightful suggestion regarding the Y15 interaction. Following your advice, we conducted an experimental Y15F mutation to investigate the proposed hydrophobic contact. The results revealed that this mutation did not greatly alter A₃AR's ligand binding ability. Additionally, to explore the role of Y265, we performed an alanine mutation, which showed that the Y265A mutant partially retained the ability to bind CF102 (**Fig. R2**). In contrast, Y15A mutation nearly abolish the activation of both CF101 and CF102 to A₃AR, even the closest distance of Y15 to CF101 is 5.4 Å, a distance that is too far to mediate direct interaction between Y15 and CF101. Yet, Y15A mutation abolish CF101 activation of A₃AR, suggesting that the effect of this mutation on ligand binding could be the alteration of the structure integrity of the receptor rather than direct interaction between Y15 and CF101.

Fig. R2 The function of Y15^{1.35} and Y265^{7.36}.

a Y15^{1.35} formed hydrophobic interaction with Y265^{7.36} in ligand-bound A₃AR. The distance between Y15 and CF101/CF102 was shown as dash lines. **b-c** Effects of CF101 and CF102 on the A₃AR and mutants using NanoBiT association assay (**b** and **c**).

Minor:

Introduction/discussion

A reference (PMID 33472058) in which m6A, an adenosine metabolite with the modification at the N6 position (as in CF101 and CF102), is proposed to be an endogenous, A₃AR-selective ligand, should be referred in the manuscript.

Response: We thank the reviewer for the insightful comment. The suggested reference has included.

Throughout the manuscript

If the authors would like to mention the adenosine receptors besides A₃AR altogether, use “the other” instead of “other”.

Response: We thank the reviewer for the kind reminder. We have made the corresponding modification.

Line 84

“NanoBiT tether strategy” should be “HiBiT tether strategy”.

Response: We have made the corresponding modification.

Line 119

“comprosed” should be “comprised”.

Response: We have made the corresponding modification.

Lines 134-135

No figure was shown for Y265.

Response: We have added a supplementary figure 8 to show the π - π interaction between Y15 and Y265.

Line 156-157

Reference is needed for the statement “According to the reported structure-activity relationship of ligands at the A3AR”.

Response: We thank the reviewer for the comment. We have cited reference (PMID: 7932588) in our revise manuscript.

Line 440

“solubliezed” should be “solubilized”.

Response: We have made the corresponding modification.

Lines 433-453

Add details for the experimental methods for the CF102-bound A3AR.

Response: Sorry for our negligence. We have been descripted the experimental methods for CF102-bound A₃AR were similar to those used for CF101-bound A₃AR.

Line 460

“CF101” was meant to be “CF102”?

Response: We have made the corresponding modification.

Reviewer #2 (Remarks to the Author):

Cai et.al reported two structures of A3AR in the agonist-binding active state. These are the firstly reported A3 receptor structures, however, the structures of other members (A1AR, A2AAR, A2ABR) in this subfamily have been solved already, including the active state and the inactive state, so it is better to explain the significance of this receptor in detail, compare it with other receptors in this subfamily, and emphasize the necessity of gaining the A3AR structure, otherwise this work is kind of routine without intriguing scientific insights. Besides, I have two major concerns:

Response: We deeply appreciate Reviewer #2's thorough review and constructive feedback. In response to your comments, we have expanded the discussion in our manuscript to more comprehensively detail the significance of the A₃AR structures in the

context of the adenosine receptor subfamily. We have included a comparative analysis with other receptors in this subfamily, emphasizing the novel insights our A₃AR structures provide. This comparison highlights the unique aspects of A₃AR, underscoring the necessity and scientific value of elucidating its structure, especially in relation to advancing our understanding of receptor-specific drug design and signaling mechanisms. This additional context aims to clearly delineate how our work contributes novel, intriguing scientific insights beyond routine structural elucidation.

- (1) The authors stated that the ECL3 part is very important for ligand recognition. However, in the density map, the ECL3 has no direct interaction with the two agonists CF101 and CF102. It needs more experiments to support this claim. For example, the GGSGGS linker substitution could affect ligand binding or not, or an MD simulation that proves the entry of ligand needs the help of ECL3 of A₃AR.

Response: We are grateful for the insightful comment regarding the role of ECL3 in ligand recognition. To substantiate our claim, we have conducted experiments with various GS linker substitutions in ECL3. These experiments demonstrate that the length and flexibility of ECL3 significantly influence A₃AR's ligand binding capacity. Although ECL3 does not directly interact with the agonists CF101 and CF102, our findings indicate that its structural characteristics, particularly its length and rigidity, play a crucial role in facilitating ligand binding to A₃AR.

- (2) The overall experimental design for validating the mutant effect is only the NanoBit assay. It is better that use multiple assays to cross-confirm the conclusions in this paper. For example, the cAMP assay, BRET assay, SPR, and other classical assays.

Response: We appreciate the reviewer's suggestion to employ multiple assays for validating our findings. In addition to the NanoBiT assay, we have incorporated the cAMP accumulation assay to assess critical residues. The consistency between these two different assays reinforces the reliability of our conclusions. This multi-faceted approach ensures a more comprehensive validation of our results, enhancing the overall robustness of our study.

Reviewer #1 (Remarks to the Author):

The authors' additional experiments satisfactorily addressed the concerns that I had raised to the submitted manuscript.

Reviewer #2 (Remarks to the Author):

I have reviewed this manuscript at the first round before. After carefully reading the revised version from me and the other reviewer, I think the authors have performed additional suggested assay and fully addressed the concerns raised by the reviewers. So I think the revised version should be suitable to be accepted in Nature Communications.

Reviewer #1 (Remarks to the Author):

The authors' additional experiments satisfactorily addressed the concerns that I had raised to the submitted manuscript.

Response: We thanks the reviewer for the previous comments aimed at enhancing the quality of our manuscript.

Reviewer #2 (Remarks to the Author):

I have reviewed this manuscript at the first round before. After carefully reading the revised version from me and the other reviewer, I think the authors have performed additional suggested assay and fully addressed the concerns raised by the reviewers. So I think the revised version should be suitable to be accepted in Nature Communications.

Response: We appreciate the insightful comments provided by the reviewer, which have significantly contributed to the improvement of our work.